# Unlocking the Antidiabetic Potential of CBD: In Vivo Preclinical Studies

**DOI:** 10.3390/ph18040446

**Published:** 2025-03-21

**Authors:** Elena Rafailovska, Elona Xhemaili, Zorica Naumovska, Olga Gigopulu, Biljana Miova, Ljubica Suturkova, Gjoshe Stefkov

**Affiliations:** 1Institute of Biology, Faculty of Natural Sciences and Mathematics, Ss. Cyril and Methodius University in Skopje, Arhimedova 3, 1000 Skopje, North Macedonia; elena.rafailovska@pmf.ukim.mk (E.R.); bmiova@pmf.ukim.mk (B.M.); 2Department of Pharmacy, Faculty of Medical Sciences, University of Tetovo, Ilinden bb, 1200 Tetovo, North Macedonia; xhemailielona@gmail.com; 3Faculty of Pharmacy, Ss. Cyril and Methodius University in Skopje, Majka Tereza 47, 1000 Skopje, North Macedonia; zose@ff.ukim.edu.mk (Z.N.); olga.gigopulu@ff.ukim.edu.mk (O.G.); ljubica.suturkova@ff.ukim.edu.mk (L.S.)

**Keywords:** cannabidiol, diabetes mellitus, insulin sensitivity, hepatic carbohydrate metabolism, laboratory animals, OGTT

## Abstract

**Background/Objectives**: Diabetes mellitus is a growing global health concern, driving the exploration of new therapies like cannabidiol (CBD), which shows potential in improving insulin sensitivity and glycemic control, though its effects on glucose metabolism remain unclear. This study evaluates CBD’s dose-dependent effects on glycemia, insulin, and hepatic carbohydrate metabolism in diabetic rats. **Methods**: The Oral Glucose Tolerance Test (OGTT) was performed in healthy rats to compare intragastric vs. intraperitoneal CBD (0.5, 5, 50 mg/kg). Diabetic rats were treated with intragastric CBD (25, 50, 100 mg/kg) or metformin (70 mg/kg) for 8 days. Blood glucose, insulin, lipid profiles, and key carbohydrate-metabolizing enzymes were analyzed. **Results**: In the OGTT, intragastric CBD reduced glycemic AUC, with 50 mg/kg showing the strongest effect, while intraperitoneal CBD had no impact. In diabetic rats, metformin and 25 mg/kg CBD lowered blood glucose, but only CBD increased insulin. The 50 mg/kg dose caused the greatest glucose reduction and moderate insulin rise, while 100 mg/kg had no effect. At 25 mg/kg, CBD inhibited glucose-6-phosphatase and increased glucose-6-phosphate. The 50 mg/kg dose further suppressed gluconeogenic enzymes, reduced glycogen phosphorylase and liver glucose, and enhanced glucose-6-phosphate, showing the strongest metabolic effects. The 100 mg/kg dose increased hexokinase but had weaker metabolic effects. Metformin improved glucose utilization and glycogen storage. CBD at 25 and 50 mg/kg reduced triacylglycerols and increased HDL, while 100 mg/kg had no effect. **Conclusions:** This study provides strong evidence of CBD’s antidiabetic potential, especially at 50 mg/kg, particularly through its modulation of glucose metabolism and tendency to regulate insulin levels.

## 1. Introduction

*Cannabis sativa* has garnered significant attention over the past 50 years due to its pharmacological properties and potential therapeutic applications [1]. The plant contains approximately 750 compounds, including the major phytocannabinoids tetrahydrocannabinol (THC) and cannabidiol (CBD) [2]. While Δ9-THC is well known for its psychoactive effects [3,4], CBD, which is primarily derived from the flowers and leaves of *C. sativa* and also present in smaller amounts in other Cannabis species [2], lacks such properties and has gained widespread use in medical contexts, including therapeutic applications and clinical trials in Europe [5].

Several medical applications of *C. sativa* are supported by reliable clinical data, including its use in managing chronic pain, multiple sclerosis, refractory epilepsy, chemotherapy-induced nausea and vomiting, appetite and weight loss associated with anxiety and sleep disorders, PTSD, and schizophrenia [6,7]. Beyond these well-documented effects, both *C. sativa* extracts and its active compounds, such as CBD and THC, have shown additional therapeutic benefits in in vitro and in vivo studies, including reductions in blood pressure, triglycerides, and cholesterol, along with antioxidant properties [6]. Notably, recent research has expanded to explore the role of CBD in metabolic disorders, particularly diabetes mellitus (DM) [8]. DM is a major global health concern, with its prevalence steadily increasing. According to the International Diabetes Federation (IDF), approximately 537 million adults were living with diabetes in 2021, a number projected to rise to 783 million by 2045 [9]. Moreover, the World Health Organization (WHO) estimates that DM-related mortality will increase by 50% by 2030, further underscoring the urgent need for effective treatment strategies [10]. Given these alarming statistics, there is a growing interest in exploring alternative therapies that may complement conventional diabetes treatments.

CBD has emerged as a potential adjunct therapy for diabetes management due to its anti-inflammatory, antioxidant, and insulin-sensitizing properties. Even though clinical trials on CBD are still limited, a Phase I study investigating the efficacy of a sublingual CBD/THC regimen (10:1) in type 2 diabetes patients found significant improvements in fasting blood sugar, HbA1c, and 2 h Oral Glucose Tolerance Test (OGTT) values, along with reduced insulin secretion and HOMA2-IR values, suggesting that CBD may enhance insulin sensitivity and glycemic control [11]. Moreover, unlike some antidiabetic treatments, CBD does not appear to be associated with an increased prevalence of liver enzyme elevation indicative of hepatic damage [12]. Preclinical studies indicate that CBD improves insulin sensitivity in both in vitro [13] and in vivo models, primarily by reducing inflammation, which plays a key role in insulin resistance [5]. In diabetic animal models, CBD has been shown to enhance glycemic control, and its effects appear to be mediated via CB2 receptor signaling [14]. Some research suggests that CBD lowers fasting and postprandial blood glucose [14] and inhibits α-glucosidase, an enzyme involved in carbohydrate digestion [15]. Furthermore, CBD may protect pancreatic β-cells from autoimmune destruction in Type 1 diabetes models, leading to improved glucose homeostasis [16]. Studies also suggest that CBD treatment ameliorates metabolic dysfunctions such as hyperglycemia and increases insulin levels in diabetic rats [14]. While these findings are promising, the precise mechanisms by which CBD modulates endogenous glucose metabolism remain unclear.

Dysregulation of hepatic glucose metabolism is a hallmark of diabetes, affecting processes such as gluconeogenesis, glycogenolysis, and glycolysis [17,18]. The liver plays a central role in maintaining glucose homeostasis, and alterations in key carbohydrate-metabolizing enzymes contribute to diabetic pathophysiology [18]. CBD has been shown to decrease the expression of genes responsible for glucose metabolism, glycolysis, and gluconeogenesis in vitro [19]. However, limited studies have investigated CBD’s direct effects on enzyme activity in hepatic carbohydrate metabolism.

Moreover, several in vivo studies have shown mixed effects on glycemic measures [20,21,22,23], or found no improvement in blood glucose levels at all [24,25]. These inconsistencies are mainly due to the different routes of administration of CBD (intraperitoneal or oral) and the different doses used in the studies (ranging from 10 mg/kg to 60 mg/kg body weight) [20,21,23]. Thus, there is a need to establish the most favorable route of administration and assess the dose-dependent effects of CBD.

The present study aims to examine the route of administration and evaluate the dose-dependent effects of CBD on glycemia. Furthermore, it assesses alterations in glucose and insulin levels to determine the potential regulatory role of CBD in maintaining glucose homeostasis. In addition, the study investigates the impact of CBD on hepatic carbohydrate metabolism in diabetic rats, focusing on its dose-dependent effects on key metabolic pathways, including gluconeogenesis, glycolysis, and glycogen metabolism, through its effects on enzyme activities and substrate concentrations.

## 2. Results

### 2.1. Oral Glucose Tolerance Test (OGTT)—Intragastric vs. Intraperitoneal Application of CBD

During the OGTT measurements in healthy rats receiving CBD treatment intragastically (Figure 1A), glucose levels peaked at similar levels across all three groups 15 min after glucose administration. Notably, the 50 mg/kg CBD group exhibited a lesser glucose increasement compared to controls and metformin-treated groups during this period. At 60 min post-administration, the 0.5 mg/kg and 50 mg/kg i.g. CBD groups, akin to the metformin-treated group, reached glucose levels comparable to controls. However, the 5 mg/kg CBD group took 90 min to reach control values. The AUC analysis of the OGTT in rats receiving CBD intragastrically (Figure 1B) revealed that the 50 mg/kg CBD treatment resulted in a significantly lower glycemic AUC compared to the other groups. Specifically, the AUC in the 50 mg/kg CBD group was approximately 10% lower than in the metformin-treated group (Met:CBD 50, *p* < 0.05), and it was also 10% and 15% lower than the AUC values in the 0.5 mg/kg and 5 mg/kg CBD groups (CBD 0.5:CBD 50, *p* < 0.05; CBD 5:CBD 50, *p* < 0.05), respectively.

The results of the OGTT performed on healthy rats receiving CBD treatment intraperitoneally (Figure 1C) showed that both metformin- and CBD-treated groups have similar dynamics regarding blood glucose levels during 120 min measurement. Namely, glucose levels in all groups peaked at similar levels 15 min after glucose administration. Similar to the metformin-treated group, the CBD-treated groups with the doses 0.5 mg/kg and 50 mg/kg reached the values of control rats 60 min after the glucose administration, while the rats treated with 5 mg/kg CBD reached control values 90 min after the glucose administration. The AUC analysis of the OGTT following intraperitoneal CBD administration (Figure 1D) revealed that treatment with 0.5 mg/kg and 5 mg/kg CBD resulted in significantly higher AUC values, compared to the control (C:CBD 0.5, *p* < 0.05; C:CBD5, *p* < 0.05) and metformin-treated group (Met:CBD 0.5, *p* < 0.05; Met:CBD5, *p* < 0.05). However, no significant differences were observed between the different CBD doses.

### 2.2. Daily Glucose Regulation During Multiple-Dose Study

The results of blood glucose levels (A) and the area under the curve (AUC) of blood glucose levels (B), measured daily during the multiple-dose study in streptozotocin-induced diabetic rats, are shown in Figure 2. The AUC results indicate that the untreated diabetic group consistently exhibited significantly higher blood glucose levels compared to the control group (C:D, *p* < 0.05). Notably, both the metformin-treated group and the CBD-treated group receiving 25 mg/kg body weight showed similar effects, significantly reducing the AUC by approximately 10% compared to the diabetic group (D:Met, *p* < 0.05; D:CBD25, *p* < 0.05; Met:CBD25, ns). Administration of CBD at a dose of 50 mg/kg body weight resulted in the most pronounced reduction, lowering the AUC by 20% (D:CBD50, *p* < 0.05). This dose also yielded a significantly lower AUC compared to the metformin-treated group (Met:CBD50, *p* < 0.05). In contrast, treatment with CBD at a dose of 100 mg/kg body weight did not produce a significant change in the AUC compared to the diabetic group (D:CBD100, ns). This treatment also showed significantly higher AUC values compared to the Met, CBD25, and CBD50 groups, respectively.

### 2.3. Hourly Profile of Blood Glucose Levels During Multiple-Dose Study on Days 2 and 6

Figure 3 depicts the hourly change in blood glucose concentration (Figure 3A,C) and area under the curve of blood glucose (Figure 3B,D) monitored on the second and sixth day of the multiple-dose study in streptozotocin-induced diabetic rats, respectively.

The AUC results showed that untreated diabetic rats consistently exhibited higher blood glucose levels on days 2 and 6 compared to the control group (Figure 3B, C:D, *p* < 0.05; Figure 3D, C:D, *p* < 0.05). On the second day, the CBD50 group demonstrated the most pronounced reduction in glycemic AUC values, with a 27.6% decrease in the first 8 h (D:CBD50, *p* < 0.05) (Figure 3B). Although this reduction was greater than that observed in the metformin group (−16.9%), the difference between the two treatments was not statistically significant (Met:CBD50, ns) (Figure 3B). In contrast, the glycemic AUC values in the CBD25 and CBD100 groups did not differ significantly from those of the untreated diabetic group (D:CBD25, D:CBD100, ns). By the sixth day, metformin exhibited a more substantial effect, reducing the glycemic AUC value by 31.2% (D:Met, *p* < 0.05), while the CBD50 group showed a more modest reduction of 12.1% (D:CBD50, *p* < 0.05) (Figure 3D). The effect of metformin on the 6th day compared to CBD50 was significantly more pronounced (Met:CBD50, *p* < 0.05). Meanwhile, the glycemic AUC values in the CBD25 and CBD100 groups remained unchanged compared to the untreated diabetic group (D:CBD25, D:CBD100, ns).

### 2.4. Blood Glucose and Insulin Concentration

Figure 4 illustrates the alterations in blood glucose (A) and insulin concentration (B) observed in both healthy and diabetic rats following treatment with three different concentrations of CBD oil and metformin during the multiple-dose study in streptozotocin-induced diabetic rats. A notable reduction in insulin concentration and high blood glucose levels were evident in the untreated diabetic group (C:D, *p* < 0.05). Metformin administration in diabetic rats did not yield significant changes compared to the diabetic control group (D:Met, ns). Treatment with CBD oil at a dose of 25 mg/kg resulted in a slight yet significant reduction in blood glucose levels (D:CBD25,−21%, *p* < 0.05), alongside a noteworthy increase in insulin concentration by 45% compared to diabetic rats (D:CBD25, *p* < 0.05). Most notably, a substantial 67% decrease in blood glucose levels (D:CBD50, *p* < 0.05) and a significant 20% increase in insulin levels were observed in the CBD50 group. Contrariwise, the treatment with a dosage of 100 mg/kg b.w. CBD did not lead to significant alterations in blood glucose nor insulin concentration compared to the diabetic control group.

### 2.5. Carbohydrate Metabolism in the Liver

Figure 5 depicts the changes in enzyme activities and substrate concentrations of carbohydrate metabolism in the liver of healthy and diabetic rats following treatment with three different doses of CBD oil and metformin during the multiple-dose study in streptozotocin-induced diabetic rats.

Compared to control animals, experimental diabetes resulted in significant increases in glucose-6-phosphatase activity (Figure 5A), alongside significant decreases in hexokinase activity (Figure 5B) and glucose-6-phosphate concentration (Figure 5C). Moreover, there were significant rises in glucose concentration (Figure 5D) and the activities of glycogen phosphorylase (Figure 5E) and fructose-1,6-bisphosphatase (Figure 5F), coupled with notable decreases in glycogen content (Figure 5G).

Metformin treatment did not induce significant changes in the activity of glucose-6-phosphatase (Figure 5A), glycogen phosphorylase (Figure 5E), or fructose-1,6-bisphosphatase (Figure 5F). However, it significantly increased hexokinase activity (Figure 5B), glucose-6-phosphate concentration (Figure 5C), and glycogen concentration (Figure 5G), while also significantly decreasing liver glucose levels (Figure 5D) compared to diabetic rats.

CBD treatment at a dose of 25 mg/kg led to a significant decrease in glucose-6-phosphatase activity (Figure 5A), with no noticeable effect on hexokinase activity (Figure 5B). It resulted in an increase in glucose-6-phosphate concentration (Figure 5C) but did not significantly affect liver glucose concentration (Figure 5D), glycogen phosphorylase activity (Figure 5E), fructose-1,6-bisphosphatase activity (Figure 5F), or glycogen concentration (Figure 5G) compared to diabetic rats.

Compared to diabetic rats, CBD treatment at a dose of 50 mg/kg resulted in a significant decrease in glucose-6-phosphatase activity (Figure 5A) while having no noticeable effect on hexokinase activity (Figure 5B). Additionally, it significantly increased glucose-6-phosphate concentration (Figure 5C) and led to a significant decrease in liver glucose concentration (Figure 5D), glycogen phosphorylase activity (Figure 5E), and fructose-1,6-bisphosphatase activity (Figure 5F), though it did not alter liver glycogen concentration (Figure 5G). The activities of glycogen phosphorylase and fructose-1,6-bisphosphatase, as well as hepatic glucose concentration in diabetic rats treated with CBD at a dose of 50 mg/kg, were significantly lower compared to diabetic rats treated with CBD at a dose of 25 mg/kg (Figure 5D,G).

Compared to diabetic rats, treatment with CBD at a dose of 100 mg/kg did not alter glucose-6-phosphatase activity (Figure 5A) or glycogen content (Figure 5G). However, it notably increased hexokinase activity (Figure 5B) and glucose-6-phosphate concentration (Figure 5C), while significantly reducing liver glucose concentration (Figure 5D), glycogen phosphorylase activity (Figure 5E), and fructose-1,6-bisphosphatase activity (Figure 5F). Additionally, glucose concentration, glycogen phosphorylase activity, and fructose-1,6-bisphosphatase activity in the CBD100 group were significantly lower compared to the CBD25 group, whereas the activities of glucose-6-phosphatase and hexokinase in the CBD100 group were significantly higher compared to both the CBD25 and CBD50 groups.

### 2.6. Blood Lipid Parameters

Figure 6 illustrates the changes in triacylglycerol, total cholesterol, and HDL levels in the blood of healthy and diabetic rats following treatment with three different doses of CBD oil (25 mg/kg, 50 mg/kg, and 100 mg/kg) and metformin during the multiple-dose study in streptozotocin-induced diabetic rats.

Diabetic rats exhibited a marked increase in triacylglycerol (TAG) levels compared to controls (C:D, *p* < 0.05), while total cholesterol and HDL levels remained unchanged (C:D, ns). Compared to diabetic rats, metformin treatment led to a significant decrease in TAG levels (D:Met, *p* < 0.05), without affecting total cholesterol and HDL levels (D:Met, ns). However, HDL levels in metformin-treated rats were significantly higher compared to healthy controls (C:Met, *p* < 0.05).

CBD treatment at a dose of 25 mg/kg led to a significant decrease in TAG levels, alongside a significant increase in total cholesterol levels compared to diabetic rats (D:CBD25, *p* < 0.05). Additionally, total cholesterol and HDL levels in the CBD25 group were significantly higher compared to healthy controls (C:CBD25, *p* < 0.05). Similarly, treatment with 50 mg/kg CBD significantly lowered TAG levels compared to diabetic rats (D:CBD50, *p* < 0.05). Total cholesterol and HDL concentrations in the CBD50 group significantly exceeded control values (C:CBD50, *p* < 0.05). In contrast, treatment with 100 mg/kg CBD had no significant effect on TAG levels (D:CBD100, ns), but significantly increased total cholesterol and HDL levels compared to diabetic rats (D:CBD100, *p* < 0.05).

## 3. Discussion

In addition to its known pharmacological properties and potential therapeutic benefits in managing metabolic syndrome and diabetes, this study provides compelling evidence supporting the antihyperglycemic and antidiabetic effects of cannabidiol (CBD) and provides insights into its mechanisms of action. Our results demonstrate the strong antidiabetic potential of CBD in diabetic animal models, particularly through its modulation of blood glucose and insulin levels as well as hepatic enzymes involved in carbohydrate metabolism.

The results from the Oral Glucose Tolerance Test (OGTT) in this study highlight the importance of optimizing CBD administration methods to enhance its efficacy in glucose regulation. The doses (0.5 mg/kg, 5 mg/kg, and 50 mg/kg) were selected based on previous studies showing glucose-lowering effects within the 5–30 mg/kg range [26,27,28]. Our findings demonstrate that intragastrically administered CBD at the highest dose (50 mg/kg) significantly improved glucose regulation, while lower doses (0.5 mg/kg and 5 mg/kg) did not produce similar effects, indicating a dose-dependent response. In contrast, intraperitoneal CBD administration did not result in significant differences in the glycemic area under the curve (AUC) across doses, suggesting that the route of administration plays a key role in CBD’s bioavailability and efficacy. This lack of effect with intraperitoneal administration aligns with findings from other studies [23]. One explanation for the better outcomes with intragastrically administered CBD is its lipophilic nature, which allows for more efficient absorption through the gastrointestinal (GI) tract and accumulation in highly vascularized tissues like the liver, essential for glucose metabolism [29,30]. Despite low oral bioavailability (around 6%) [31,32], the absorption via the GI tract, combined with CBD’s protein-binding capacity and long half-life, may enhance its therapeutic effects when administered at appropriate doses [33,34].

Based on the results of the OGTT, we selected intragastric administration for the STZ-induced diabetes experiment. The doses of 25 mg/kg, 50 mg/kg, and 100 mg/kg were chosen based on previous studies demonstrating the beneficial effects of CBD in diabetic and insulin resistance models. Namely, studies using doses between 10 mg/kg and 30 mg/kg have shown improvements in glycemic control and insulin sensitivity [1,27,35,36]. The inclusion of higher doses aimed to explore potential enhanced therapeutic effects and assess whether increasing concentrations would yield stronger metabolic benefits or reveal a saturation threshold.

Throughout the 8-day treatment period, the administration of 50 mg/kg CBD consistently showed the most significant reduction in daily blood glucose levels among diabetic rats, lowering blood glucose by approximately 20%. Notably, this effect was more pronounced than that observed with metformin, a standard antidiabetic treatment. Interestingly, the 100 mg/kg dose did not provide additional benefits and was less effective than the lower doses. This inverted-U dose–response relationship aligns with previous research on cannabinoids, where biphasic effects have been reported for different pathologies [37,38]. At higher doses, CBD’s efficacy may be reduced due to receptor saturation or receptor desensitization. Chronic activation of CB1 receptors, which play a key role in energy homeostasis [39], can trigger feedback processes that counteract initial receptor activity [40], potentially explaining the diminished therapeutic effects observed at increased CBD dosages. Additionally, CBD interacts with multiple receptors, including transient receptor potential (TRP) channels, particularly TRPV1 [41], which is involved in glucose homeostasis [42]. Activation of TRPV1 can initially enhance signaling but may subsequently lead to a prolonged refractory state, commonly referred to as desensitization [43]. This desensitization could contribute to the reduced efficacy observed at higher CBD doses.

Hourly glycemic profiling further supported the efficacy of the 50 mg/kg dose, which reduced blood glucose levels by 27.6% within 8 h on the second day of treatment, comparable to the effect of metformin. However, by the sixth day of treatment, metformin exhibited superior glycemic control (31.2% reduction) compared to CBD50 (12.1% reduction). These findings suggest that while CBD is effective in providing short-term glycemic control, metformin offers more sustained benefits over longer treatment periods. Metformin’s prolonged efficacy could be attributed to its well-established mechanisms of action, including the cumulative reduction in hepatic glucose production [44], activation of AMP-activated protein kinase (AMPK), and modulation of cellular redox balance [45], all of which collectively contribute to stable blood sugar levels over time. In contrast, the transient nature of CBD’s efficacy raises the possibility of combination therapies to leverage the rapid effects of CBD alongside metformin’s sustained benefits.

This study also highlights the dose-dependent impact of CBD on glycemic control and insulin levels, with significant changes observed at lower doses (25 and 50 mg/kg), while the highest dose (100 mg/kg) exhibited minimal effects. Previous studies have demonstrated CBD’s potential in modulating glucose homeostasis [14,26,36,46], and the dose-dependent effects observed here underscore the complexity of its action. In vivo studies have shown that CBD reduces hyperglycemia and increases plasma insulin levels at doses ranging from 5 mg/kg to 30 mg/kg [28,46]. However, starting at 60 mg/kg, no improvement in glycemic measures was observed in male Wistar rats [23], and similarly, no changes in blood glucose or serum insulin were noted in humans after 13 weeks of daily oral administration of 200 mg CBD [20]. This lack of effect at higher doses may suggest a saturation point beyond which CBD fails to enhance glycemic control. As previously speculated, at higher doses, CBD may lose its efficacy in glycemic control due to receptor desensitization, activation of counter-regulatory mechanisms, and non-specific interactions. Moreover, CBD at higher doses may interact with alternative molecular targets, leading to metabolic outcomes that counteract its glucose-lowering effects [47]. While we focused on glycemic control and insulin levels, future studies on insulin sensitivity using the Homeostatic Model Assessment of Insulin Resistance (HOMA-IR) could enhance the translational value of these findings.

The findings of this study further emphasize the liver’s pivotal role in mediating CBD’s antidiabetic effects, given its centrality in glucose metabolism. CBD’s influence on hepatic gluconeogenesis, glycogenolysis, and glycolysis may account for the observed improvements in glycemic control. In this sense, experimental diabetes significantly altered the activities of enzymes and substrates involved in carbohydrate metabolism. Our findings indicate that experimental diabetes increased gluconeogenesis, as evidenced by the elevated activities of glucose-6-phosphatase and fructose-1,6-bisphosphatase, likely due to decreased insulin levels in diabetic rats. Since insulin directly suppresses gluconeogenic enzymes [48], its deficiency contributes to this upregulation. Simultaneously, the lack of insulin suppresses the activity of enzymes associated with glucose metabolism [49], such as hexokinase (HK), as demonstrated in our results. The reduced activity of hexokinase in diabetic rats accounts for the decrease in glucose-6-phosphate (G6P) concentration, as HK is responsible for glucose phosphorylation and subsequent G6P production [50]. The diabetic state also lowered the glycogen content in the liver, probably due to diminished glucose uptake in hepatocytes [51] and reduced activity in glycogen synthetase in response to decreased insulin levels [52]. Additionally, we observed lower activity of glycogen phosphorylase in the liver of diabetic animals, likely due to reduced glycogen stores, as phosphorylase is inactivated when glycogen levels are low to minimize glycogenolysis [53].

The treatment of diabetic animals with CBD at a dose of 25 mg/kg selectively modulated certain aspects of carbohydrate metabolism, particularly by lowering the activity of the gluconeogenic enzyme glucose-6-phosphatase. The increase in glucose-6-phosphate concentration is consistent with the decreased activity of glucose-6-phosphatase. Since less glucose-6-phosphate is being converted back to glucose, its intracellular concentration rises [54]. These changes support the slight glycemic control observed in this experimental group.

The treatment with CBD at dose of 50 mg/kg exerts a stronger inhibitory effect on gluconeogenesis and glycogenolysis compared to the lower dose. Treatment with this concentration leads to a notable decrease in glucose-6-phosphatase and fructose-1,6-bisphosphatase activities, reflecting a robust inhibition of gluconeogenesis [55]. This effect is further supported by the significant reduction in liver glucose concentration, indicating that 50 mg/kg CBD effectively reduces hepatic glucose production [56]. These findings align with previous literature demonstrating that fructose-1,6-bisphosphatase and glucose-6-phosphatase activities are suppressed in vitro, in a dose-dependent manner, when incubated with cannabidiol. Specifically, doses ranging from 120 to 240 µg/mL of CBD have been shown to inhibit these key gluconeogenic enzymes [57]. Additionally, CBD has been reported to decrease the expression of genes involved in gluconeogenesis in vitro [19], further supporting our observed suppression of glucose-6-phosphatase at 25 mg/kg and the stronger inhibition at 50 mg/kg. The observed decrease in gluconeogenesis and glycogenolysis might be due to the modulation of gene expression by CBD; however, further studies are needed to clarify its precise mechanism of action. Additionally, the treatment at a dose of 50 mg/kg reduces glycogen phosphorylase activity, indicating a suppression of glycogenolysis, which helps to preserve glycogen stores. This is consistent with reports that cannabidiol suppresses glycogen phosphorylase activity in a dose-dependent manner [57]. Despite this suppression, in our results, liver glycogen concentration remains unchanged, suggesting that while glycogen breakdown is inhibited, glycogen synthesis may still be impaired [58]. Furthermore, we observed non-significant changes in hexokinase activity, despite the significant increase in glucose-6-phosphate concentration, which might be explained by feedback inhibition of hexokinase by high levels of glucose-6-phosphate [59]. Interestingly, the literature indicates that CBD significantly upregulates mRNA expression of hexokinases [60], suggesting that CBD’s effects on hexokinase may involve regulatory mechanisms at the transcriptional level, which may not immediately translate into changes in enzyme activity.

The effects of CBD treatment at a dose of 100 mg/kg present a distinct pattern compared to the 25 mg/kg and 50 mg/kg doses, revealing different impacts on hepatic carbohydrate metabolism in diabetic rats. The 100 mg/kg CBD dose did not reduce glucose-6-phosphatase activity, indicating that gluconeogenesis continues [55], which contributes to persistent hyperglycemia. Despite increased hexokinase activity and glucose-6-phosphate levels, liver glucose concentrations are lower, likely because glucose is being released into the bloodstream. Although glycogen phosphorylase activity is reduced, glycogen levels remain low, suggesting that glycogen synthesis is still impaired [58], similar to the effects observed at 50 mg/kg. Consistent with our earlier observations, this dose of CBD did not result in effective glycemic control.

The results from metformin treatment in this study showed that it did not induce significant changes in the activity of key enzymes involved in gluconeogenesis (glucose-6-phosphatase, fructose-1,6-bisphosphatase) or glycogenolysis (glycogen phosphorylase). However, metformin significantly increased hexokinase activity, glucose-6-phosphate levels, and glycogen concentration, while reducing liver glucose levels compared to diabetic rats. These findings suggest that during the 8-day treatment, metformin’s action likely focuses on improving glucose metabolism by increasing the efficiency of glucose utilization, rather than directly inhibiting glucose production [61]. The increase in hexokinase activity indicates enhanced glucose phosphorylation, facilitating glucose storage and utilization within liver cells [62]. The elevated glucose-6-phosphate concentration further supports this, as glucose-6-phosphate is a key intermediate in both glycolysis and glycogen synthesis [54]. The increased levels of glucose-6-phosphate together with lowered liver glucose levels could also indicate enhanced glucose utilization in the liver by storing glucose as glycogen [63], which subsequently results in increased glycogen levels in the liver. This pattern of results aligns with the known mechanisms of action of metformin, where it promotes glucose utilization [64]. Overall, CBD at 50 mg/kg reduces liver glucose production by inhibiting gluconeogenesis and glycogenolysis, while metformin enhances glucose utilization and storage. Notably, CBD exhibits metabolic effects similar to other natural compounds, such as berberine, curcumin, and resveratrol, which enhance glucose metabolism by reducing hepatic glucose production [65] or promoting glucose uptake [66]. However, future research is needed to address the underlying molecular mechanisms of action or long-term effects of CBD treatment, and to better compare the efficacy of CBD with metformin, which is known for its long-term benefits in glucose metabolism and insulin sensitivity [64].

CBD oil at doses of 25, 50, and 100 mg/kg significantly reduced triglyceride (TAG) levels in diabetic rats, although its efficacy in restoring TAG levels to control values was lower than that of metformin. This effect is consistent with previous findings, where CBD at doses of 10 and 30 mg/kg was shown to lower plasma triglycerides in Wistar rats [25]. The observed reduction in TAG levels may be attributed to CBD’s ability to modulate lipid metabolism through PPARγ activation, which has been implicated in improving lipid profiles in diabetic conditions [25].

Interestingly, the highest CBD dose (100 mg/kg) resulted in increased total cholesterol and HDL levels, a finding that aligns with previous studies [27,67]. This increase in HDL may suggest a potential cardioprotective role of CBD, as elevated HDL levels are typically associated with improved lipid transport and reduced cardiovascular risk [67]. Human trials have provided additional insights into the lipid-modulating effects of CBD, with CBD-rich treatments in type 2 diabetic patients leading to reductions in both plasma triglycerides and total cholesterol [11]. However, the concurrent rise in total cholesterol warrants further investigation, as it could indicate a shift in lipid metabolism rather than an overall improvement in lipid homeostasis.

This study demonstrates that orally administered CBD reduces blood glucose levels in diabetic rats by inhibiting hepatic glucose production, showing effects comparable to but shorter-lasting than metformin. These findings suggest potential therapeutic benefits for humans. CBD oil could offer advantages as an antidiabetic treatment due to its excellent safety profile and lack of liver toxicity, with human trials indicating that doses up to 1500 mg per day are well tolerated [12,68]. However, individual responses to CBD may vary, potentially leading to inconsistent effects [69].

A significant drawback of CBD is its interaction with liver enzymes, particularly those in the cytochrome P450 system, which plays a key role in drug metabolism [70]. CBD inhibits CYP2C9 and CYP3A4, enzymes responsible for metabolizing many drugs, including antidiabetic medications such as sulfonylureas and metformin [68,71]. While CBD oil shows promise as an adjunct therapy for diabetes [8], further research is needed to evaluate the effects of chronic CBD use and the potential for drug interactions before it can be widely recommended for diabetic patients.

## 4. Materials and Methods

### 4.1. Animals and Tissue Procedures

For the experimental purposes, male Wistar rats (3–4 months old with a body weight of 250–300 g) were used. The animals were fed with standard laboratory chow and given water ad libitum. A total of 112 rats were used, consisting of 64 rats for the OGTT experiment (8 rats per group across 8 groups) and 48 rats for the multiple-dose experiment in the streptozotocin-induced diabetes experiment (8 rats per group across 6 groups).

During the whole experimental period, they were housed under a 12-hour light regime (6 a.m.–6 p.m. light). The animals were always sacrificed at the same time of the day (9–10 a.m.), with a standard laparotomic procedure, after induction of anesthesia with Na-thiopental narcosis (45 mg/kg), given by the guide of the EC Directive 86/609/EEC.

All experiments were performed in accordance with the current ethical norms approved by the Animal Ethics Committee of the University “Ss. Cyril and Methodius”, Skopje, R. North Macedonia (License number 10-409/6 from 10-03-2024), following the Directive 2010/63/EU for the protection of animals used for scientific purposes of the European Commission.

### 4.2. Treatment of Animals

A commercially available 20% CBD oil, suspended in olive oil (Replek Farm Ltd., Skopje, North Macedonia), was purchased from a local pharmacy and used in this study. As a reference antidiabetic agent (positive control), metformin hydrochloride was used in the form of a secondary reference standard with a molecular weight of 165.62 g/mol and a purity of 99.89% (Alkaloid AD, Skopje, North Macedonia). Metformin was dissolved in distilled water and administered at a dose of 70 mg/kg body weight [72].

### 4.3. Oral Glucose Tolerance Test (OGTT)

To evaluate the hypoglycemic effect of CBD oil, an experiment was conducted using healthy male and female rats. The CBD oil used in the study was suspended in olive oil and administered in three doses: 0.5 mg/kg, 5 mg/kg, and 50 mg/kg. CBD oil treatments were applied using two different administration routes, intragastric (i.g.) and intraperitoneal (i.p.), to determine the more effective method of delivery.

The experiment aimed to assess glycemic regulation following a single administration of metformin or CBD oil at three different concentrations (0.5 mg/kg, 5 mg/kg, and 50 mg/kg). The rats were divided into eight experimental groups, 8 rats per group (n = 64), which are presented in Table 1.

Glycemia was measured in all experimental groups using a glucometer, with blood samples collected from the tail vein immediately before CBD oil administration. Thirty minutes after treatment, all animals received a glucose solution (2 g/kg body weight) via an intragastric tube [73,74]. Blood glucose levels were subsequently measured at 0, 15, 30, 60, 90, and 120 min after glucose administration in the same animals at each time-point. Blood samples were taken from the tail with great care to minimize stress and ensure accuracy in glucose readings. Six hours prior to treatment, food was withheld from all experimental groups, and animals remained fasted throughout the measurement period, while water was available ad libitum [75].

### 4.4. Multiple-Dose Study in Streptozotocin-Induced Diabetic Rats

Streptozotocin (STZ)-induced diabetes was established by a single intraperitoneal injection of STZ (45 mg/kg body weight), freshly dissolved in 0.1 M citrate buffer (pH 4.5) [75]. For experimental purposes, only animals that exhibited clear diabetic symptoms (fasting blood glucose levels above 15 mmol/L) 24–48 h after diabetes induction were included in the study [14,75].

To investigate the effect of an 8-day treatment with different doses of CBD, the rats were randomly divided into the following groups, with 8 rats per group (n = 48), as presented in Table 2.

All treatments were administered once daily to 8 h fasted rats (with water available ad libitum). After treatment, the rats were given free access to food and water. Fasting blood glucose was measured before treatment initiation using a glucometer, with blood samples collected from the tail vein. During the 8-day treatment period, blood glucose levels were also assessed on days 2 and 6 of the experiment at 0h (baseline) and at 1, 2, 4, and 8 h post-administration. At the end of the experiment, the rats were sacrificed, and blood samples were collected from the dorsal vein into serum and plasma separation tubes. After centrifugation, the resulting supernatant, designated as serum or plasma, was transferred into clean polypropylene tubes and stored at 2–8 °C until further analysis. The rats’ livers were isolated, washed with a cold saline solution, and immediately immersed in liquid nitrogen. Tissue powder was prepared at liquid nitrogen temperature, and portions were kept at −80 °C. Before analysis, the tissue powder was homogenized with an ultrasonic homogenizer (Cole-Parmer Instrument-4710, USA/Illionis) for 7–10 s, maintaining a temperature of 0–4 °C (on ice) throughout the process.

### 4.5. Biochemical Analysis

Blood glucose was measured using the GOD method with a colorimetric kit (Human, Weisbaden, Germany). Plasma insulin levels were assessed using a commercially available enzyme-linked immunosorbent assay (ELISA) kit (Mercodia AB, Uppsala, Sweden). Additionally, serum levels of triglycerides (TAGs), total cholesterol (TC), and high-density lipoprotein (HDL) were evaluated using commercial enzymatic tests (Human, Germany).

### 4.6. Enzymes Activities and Substrates Concentration

All the enzymes’ activities and substrates concentrations were measured in liver homogenates, prepared in appropriate homogenization media. Liver glycogen, glucose, and glucose-6-phosphate concentrations [76] were determined in perchlorate homogenates, neutralized with 5M K_2_CO_3_, by production of NADPH at 340 nm. The substrate concentration was expressed as μmol/g tissue. Glucose-6-phosphatase (E.C. 3.1.3.9) was assayed by the method of Hers [77], while the fructose-1,6-bisphosphatase (E.C. 3.1.3.11) activity was assayed using the method by Freedland and Harper [78]. The activity of liver glycogen phosphorylase *a* (E.C. 2.4.1.1) was determined by Stalmans et al. [79]. For determination of hexokinase (EC 2.7.1.1) activity [80], we measured the production of NADH at 340 nm. For the interpretation of enzyme activity as specific enzyme activity in the tissue extracts, the total quantity of the proteins was determined by Lowry et al. [81], by using bovine serum albumin as a standard. Enzyme activity was expressed as nmol Pi/min/mg protein. The amount of released inorganic phosphate was determined by Fiske and Subbarow [82].

### 4.7. Statistics

Results are presented as means ± SD. Normality of the data was assessed using the Shapiro–Wilk test. Since the data met the assumption of normality, one-way ANOVA with Tukey’s post hoc test was used for comparisons between groups. Statistical analysis was performed using GraphPad Prism 9. A *p*-value of <0.05 was considered statistically significant.

## 5. Conclusions

This study provides promising evidence for the potential antihyperglycemic and antidiabetic effects of cannabidiol (CBD), highlighting its ability to modulate blood glucose and insulin levels in diabetic animal models. The findings suggest that CBD’s efficacy may be dose-dependent and influenced by the route of administration, with intragastric administration at 50 mg/kg producing the most notable glucose-lowering effects. Interestingly, at this dose, CBD demonstrated greater short-term glycemic control compared to metformin; however, its effects appeared transient, whereas metformin maintained prolonged benefits. Mechanistically, CBD at 50 mg/kg was associated with the suppression of gluconeogenesis and glycogenolysis, likely through the inhibition of key enzymes such as glucose-6-phosphatase and fructose-1,6-bisphosphatase, leading to reduced hepatic glucose production. Nevertheless, higher doses (100 mg/kg) did not further enhance these metabolic effects.

While these findings provide valuable insights into CBD’s metabolic effects, further research is required to fully elucidate its long-term efficacy and mechanisms of action, particularly in chronic treatment models. Future studies should also explore potential molecular targets and signaling pathways to better understand the role of CBD in glucose homeostasis.

## Figures and Tables

**Figure 1 pharmaceuticals-18-00446-f001:**
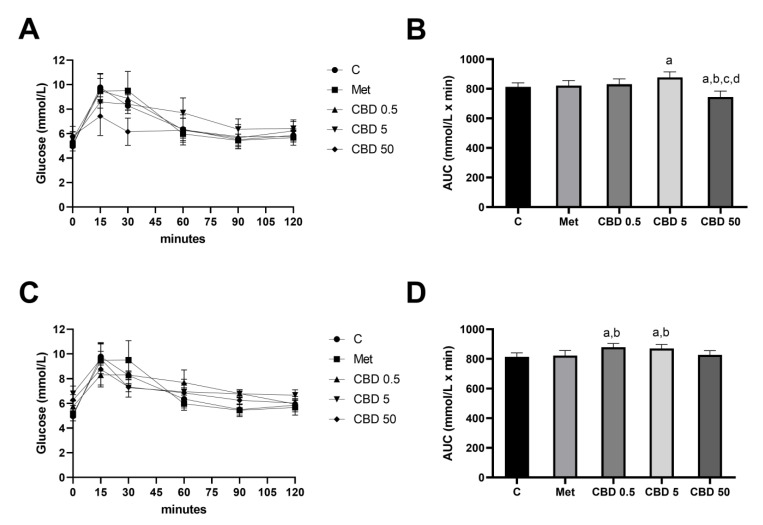
Blood glucose levels (**A**,**C**) and area under the curve (AUC) of blood glucose levels (**B**,**D**) measured during the Oral Glucose Tolerance Test (OGTT) in healthy rats treated with cannabidiol (CBD). CBD was administered in doses of 0.5 mg/kg, 5 mg/kg, and 50 mg/kg body weight either intragastrically (**A**,**B**) or intraperitoneally (**C**,**D**). Treatment groups included a control group (C), a metformin-treated group (Met, 70 mg/kg), and CBD-treated groups receiving 0.5 mg/kg (CBD 0.5), 5 mg/kg (CBD 5), or 50 mg/kg (CBD 50). Data are presented as mean ± SD. Significant differences (*p* < 0.05) are indicated as follows: a—compared to C; b—compared to Met; c—compared to CBD 0.5; d—compared to CBD 5.

**Figure 2 pharmaceuticals-18-00446-f002:**
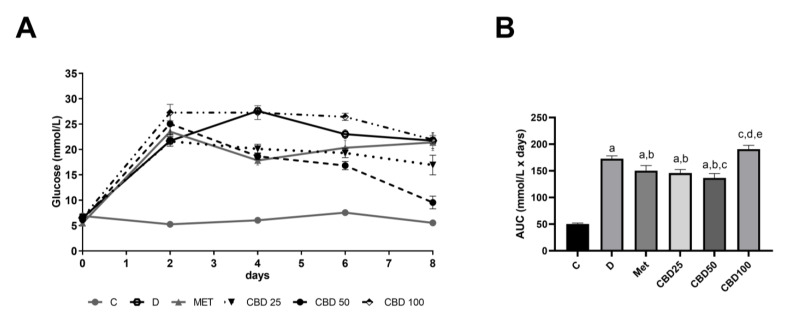
Blood glucose levels (**A**) monitored daily and area under the curve (AUC) of blood glucose levels (**B**) measured daily during the multiple-dose study in streptozotocin-induced diabetic rats. Treatment groups included a control group (C), a diabetic control group (D), a metformin-treated group (Met, 70 mg/kg), and cannabidiol (CBD) treatment groups receiving 25 mg/kg (CBD25), 50 mg/kg (CBD50), or 100 mg/kg (CBD100). Data are presented as mean ± SD. Significant differences (*p* < 0.05) are indicated as follows: a—compared to C; b—compared to D; c—compared to Met; d—compared to CBD25; e—compared to CBD50.

**Figure 3 pharmaceuticals-18-00446-f003:**
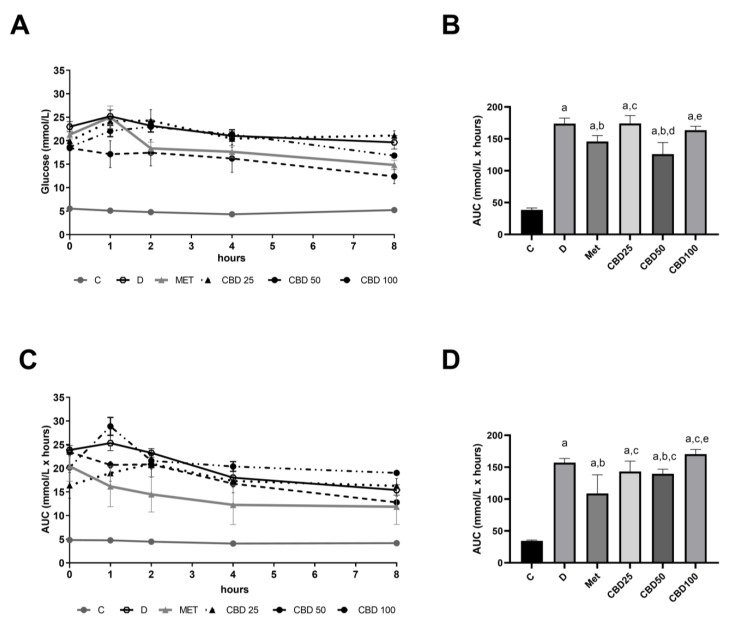
Blood glucose levels (**A**) monitored hourly, and the corresponding area under the curve (AUC) values (**B**) measured on the 2nd day; blood glucose levels (**C**) monitored hourly and the corresponding area under the curve (AUC) values (**D**) measured on the 6th day of the multiple-dose study in streptozotocin-induced diabetic rats. Treatment groups included a control group (C), a diabetic control group (D), a metformin-treated group (Met, 70 mg/kg), and cannabidiol (CBD) treatment groups receiving 25 mg/kg (CBD25), 50 mg/kg (CBD50), or 100 mg/kg (CBD100). Data are presented as mean ± SD. Significant differences (*p* < 0.05) are indicated as follows: a—compared to C; b—compared to D; c—compared to Met; d—compared to CBD25; e—compared to CBD50.

**Figure 4 pharmaceuticals-18-00446-f004:**
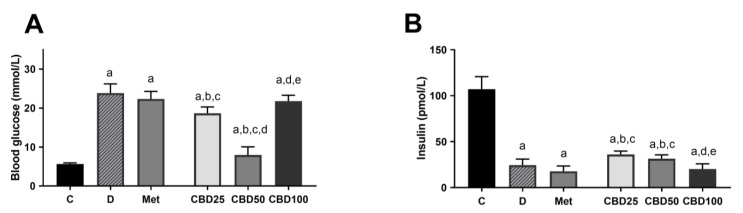
Blood glucose (**A**) and plasma insulin concentration (**B**) in healthy and STZ-induced diabetic rats treated with CBD oil at three different doses and metformin, measured at the end of the multiple-dose study. Treatment groups included a control group (C), a diabetic control group (D), a metformin-treated group (Met, 70 mg/kg), and cannabidiol (CBD) treatment groups receiving 25 mg/kg (CBD25), 50 mg/kg (CBD50), or 100 mg/kg (CBD100). Data are presented as mean ± SD. Significant differences (*p* < 0.05) are indicated as follows: a—compared to C; b—compared to D; c—compared to Met; d—compared to CBD25; e—compared to CBD50.

**Figure 5 pharmaceuticals-18-00446-f005:**
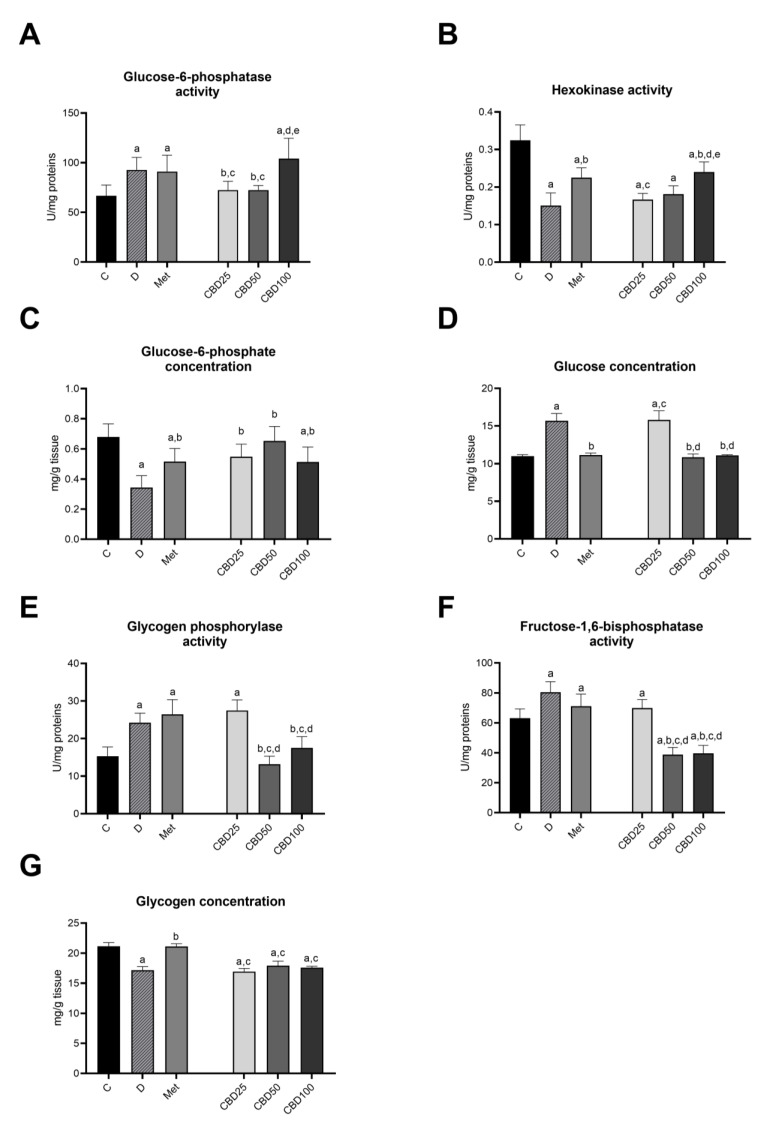
Activity of glucose-6-phosphatase (**A**), hexokinase (**B**), glucose-6-phosphate concentration (**C**), glucose concentration (**D**), glycogen phosphorylase activity (**E**), fructose-1,6-bisphosphatase activity (**F**), and glycogen concentration (**G**) in the liver of healthy and STZ-induced diabetic rats treated with CBD extracts and metformin in the multiple-dose study. Treatment groups included a control group (C), a diabetic control group (D), a metformin-treated group (Met, 70 mg/kg), and cannabidiol (CBD) treatment groups receiving 25 mg/kg (CBD25), 50 mg/kg (CBD50), or 100 mg/kg (CBD100). Data are presented as mean ± SD. Significant differences (*p* < 0.05) are indicated as follows: a—compared to C; b—compared to D; c—compared to Met; d—compared to CBD25; e—compared to CBD50.

**Figure 6 pharmaceuticals-18-00446-f006:**
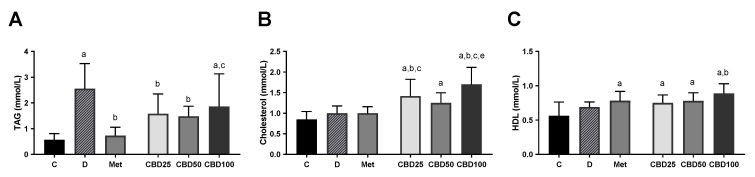
Triacylglycerol (TAG) (**A**), total cholesterol (**B**), and HDL cholesterol (**C**) levels, measured at the end of the multiple-dose study in streptozotocin-induced diabetic rats. Treatment groups included a control group (C), a diabetic control group (D), a metformin-treated group (Met, 70 mg/kg), and cannabidiol (CBD) treatment groups receiving 25 mg/kg (CBD25), 50 mg/kg (CBD50), or 100 mg/kg (CBD100). Data are presented as mean ± SD. Significant differences (*p* < 0.05) are indicated as follows: a—compared to C; b—compared to D; c—compared to Met; d—compared to CBD25; e—compared to CBD50.

**Table 1 pharmaceuticals-18-00446-t001:** Groups of healthy rats included in the Oral Glucose Tolerance Test (OGTT) study divided according to the treatment regime.

Code	Treatment	Dose
C	Water	1 mL/100 g body weight
Met	Metformin	70 mg/kg body weight
CBD 0.5 i.g.	CBD oil extract	0.5 mg/kg
CBD 5 i.g.	CBD oil extract	5 mg/kg
CBD 50 i.g.	CBD oil extract	50 mg/kg
CBD 0.5 i.p.	CBD oil extract	0.5 mg/kg
CBD 5 i.p.	CBD oil extract	5 mg/kg
CBD 50 i.p.	CBD oil extract	50 mg/kg

**Table 2 pharmaceuticals-18-00446-t002:** Groups of rats included in the multiple-dose study divided according to the treatment regime.

Code	Type of Rats	Treatment	Dose
C	Normal control	Water	1 mL/100 g body weight
D	Diabetic control	Olive oil	1 mL/100 g body weight
Met	Diabetic rats	Metformin	70 mg/kg body weight
CBD25	Diabetic rats	CBD oil extract	25 mg/kg body weight
CBD50	Diabetic rats	CBD oil extract	50 mg/kg body weight
CBD100	Diabetic rats	CBD oil extract	100 mg/kg body weight

## Data Availability

The original contributions presented in this study are included in the article. Further inquiries can be directed to the corresponding author.

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
