# Peer review of "Unlocking the Antidiabetic Potential of CBD: In Vivo Preclinical Studies"

_pharmaceuticals, 2025, doi:10.3390/ph18040446_

Round 1
Reviewer 1 Report
Comments and Suggestions for Authors
The manuscript explores the effects of cannabidiol (CBD) on glucose metabolism in a streptozotocin-induced diabetic rat model, providing valuable insights into its potential antihyperglycemic properties. The study is well-structured, employing a relevant experimental model and including a dose-response evaluation of CBD in comparison to metformin. The findings suggest that CBD, particularly at a dose of 50 mg/kg, improves glycemic control by modulating key hepatic enzymes, though further exploration of the biphasic dose-response relationship is warranted. While the study provides valuable preclinical evidence, the discussion could better address translational implications for human diabetes management and the molecular mechanisms underlying CBD’s effects. Overall, the study presents promising findings, but a major revision is necessary to strengthen its scientific rigor and validity.

The manuscript is generally well-written and uses appropriate scientific terminology. However, some areas could benefit from improved fluency and precision. There are occasional redundancies, inconsistencies in verb tense, and abrupt transitions between ideas that affect the overall readability. Additionally, certain statements could be phrased more cautiously to avoid overgeneralization of the findings. A thorough language revision would enhance clarity and coherence, ensuring a more natural and precise scientific presentation.
Reviewer 2 Report
Comments and Suggestions for Authors
The manuscript presents an interesting and relevant investigation into the potential antidiabetic effects of CBD, particularly focusing on its impact on glucose metabolism in diabetic rats. However, there are significant issues in multiple areas that need to be addressed to improve clarity, accuracy, and scientific rigor.
-
The background section lacks a focused justification for the study. While the authors discuss CBD's potential benefits in diabetes, the link to hepatic carbohydrate metabolism is not well established. Additionally, references to previous studies on CBD’s glucose metabolism effects should be more critically analyzed rather than listed as general claims.
-
The methodology section has multiple issues that compromise the reproducibility of the study.
- The criteria for selecting diabetic rats (fasting glucose >15 mmol/L) should be clearly justified with references. Also, it is unclear why only a single dose of STZ was used, given that variability in diabetes induction is common.
- The rationale for choosing 25, 50, and 100 mg/kg CBD doses is unclear. Why were these particular doses selected, and do they align with previous research or clinical relevance? The discussion of the biphasic response later in the manuscript would benefit from a more explicit justification in the methods.
- The study assumes that intraperitoneal administration is ineffective based on glucose profiles but does not provide pharmacokinetic data (e.g., plasma CBD levels). This weakens the conclusion that oral administration is superior.
- The methodology lacks details on fasting duration before the OGTT and the justification for selecting a 2 g/kg glucose dose. Additionally, it is not stated whether the same rat was used for multiple time-point measurements, which would influence stress-related glycemic fluctuations.
-
The results section includes several issues that hinder interpretation and credibility.
- The claim that CBD "surpasses the short-term efficacy of metformin" is misleading. The glucose-lowering effect of CBD (27.6% reduction at day 2) is compared to metformin (16.9% reduction) without statistical validation. Additionally, by day 6, metformin outperforms CBD50, which contradicts the earlier claim.
- Figures 1–6 lack clear legends explaining the experimental conditions, and significant differences (p-values) are not adequately marked on all graphs. The visual representation should clearly indicate comparisons between treatment groups.
- The discussion of the AUC analysis is inconsistent. For instance, in Figure 1B, the AUC for CBD50 is stated to be lower than metformin, but no direct statistical comparison is provided. Without statistical verification, these claims remain speculative.
- The statement that CBD has a "lipid-lowering effect" is misleading. While it increases HDL levels, it does not significantly reduce triglycerides compared to metformin, and at higher doses, it appears to increase total cholesterol. This contradicts the therapeutic benefit claim.
-
The discussion contains multiple speculative statements and lacks critical engagement with limitations.
- The authors mention a biphasic response but fail to discuss possible mechanisms. One plausible explanation is receptor desensitization or CBD’s interaction with different metabolic pathways at high doses. This needs to be addressed with references.
- The discussion does not contextualize CBD’s effects relative to known antidiabetic compounds beyond metformin. A comparative discussion with other natural products affecting glucose metabolism (e.g., berberine, curcumin) could strengthen the argument for CBD’s potential.
- The claim that CBD50 regulates gluconeogenesis and glycogenolysis is not sufficiently supported by mechanistic data. The authors should consider including gene expression analyses for key gluconeogenic enzymes to strengthen their conclusions.
- While the study suggests a promising role for CBD, no mention is made of potential adverse effects, such as liver enzyme alterations or long-term metabolic consequences. These aspects should be acknowledged.
- The conclusion overstates the findings. The phrase “provides strong evidence” should be softened, as this is a preclinical study with limited mechanistic data. Furthermore, the claim that CBD is a promising candidate for diabetes treatment should be revised to acknowledge the need for further validation in clinical trials.
Reviewer 3 Report
Comments and Suggestions for Authors
Overall, this article presents significant findings on the antidiabetic potential of CBD and contributes valuable knowledge to the field of metabolic disease research. I have some suggestions for improvement:
- Additional discussion on the long-term implications of CBD treatment and potential application in humans would enhance the article’s relevance to clinical applications.
- Minor editing: (1) please italicize scientific names, (2) check the font size and typo in L. 101, 112, and 194.
Reviewer 4 Report
Comments and Suggestions for Authors
Manuscript ID: pharmaceuticals-3490283
This study explores the antihyperglycemic effects of cannabidiol (CBD) oil in diabetic rats, revealing a dose-dependent response in regulating glucose metabolism. The 50 mg/kg oral dose effectively reduced blood glucose levels by 27.6%, surpassing metformin in short-term efficacy, while higher doses showed no additional benefits. Findings suggest CBD's potential in diabetes management, warranting further research for therapeutic optimization. The manuscript may be further improved by following suggestions.
- The author should include recent global statistics on diabetes mellitus (as the current information is brief), along with key challenges, research gaps, and a future roadmap, supported by appropriate references in the introduction section.
- Provide registration number with date of Animal Ethics Committee of the University “Ss. Cyril and Methodius”, 102 Skopje, R. North Macedonia.
- In section 2.2, author has mentioned “Metformin was dissolved in distilled water and administered at a dose of 70 mg/kg body weight.” Provide suitable references for dose selection. Additionally, a commercially available 20% CBD was purchased from a local pharmacy and used in this study. Can author provide details about manufacturer, quality control data? % purity? On what basis doses of CBD oil extract was decided? What is LD50 in rat?
- Section 2.5 need further detail explanation about the methodology used.
- What are the natural sources of Cannabidiol (CBD), mention in introduction part with suitable references. Is there any potential toxicity reported, highlight in introduction or discussion part?
- The study highlights the significant difference in efficacy between oral and intraperitoneal CBD administration. Can the authors provide additional discussion or references to explain the pharmacokinetic differences influencing CBD’s bioavailability and therapeutic action?
- The study reports an inverted-U dose-response, where the 100 mg/kg dose was less effective than 50 mg/kg. Can the authors elaborate on potential receptor saturation mechanisms or counter-regulatory pathways that may explain this phenomenon?
- While CBD50 showed strong short-term glycemic control, metformin exhibited more sustained benefits. Could authors provide mechanistic insights or references explaining why metformin maintains long-term efficacy better than CBD?
- CBD may regulate glucose metabolism at both enzymatic and gene expression levels. Can the authors support this claim with experimental data or references related to CBD’s impact on gene expression of gluconeogenic enzymes?
- The study notes that CBD improved HDL levels but was less effective than metformin in reducing hypertriglyceridemia. Can the authors discuss potential mechanisms by which CBD influences lipid metabolism and whether its cardiovascular benefits outweigh its limitations in triglyceride regulation?
Round 2
Reviewer 1 Report
Comments and Suggestions for Authors
Thank you for your efforts in addressing the comments and for the modifications made to the manuscript. Significant improvements have been implemented in terms of clarity, the discussion of clinical relevance, and the presentation of results. I also appreciate the explanation regarding the additional analyses you are conducting in ongoing studies. However, I believe some aspects could be further strengthened with minor revision. These minor revisions do not require experimental changes but would enhance the clarity and impact of the study. With these modifications, I believe the manuscript will be ready for publication.

Author Response
Thank you for your helpful suggestion regarding the clarification of the acronym OGTT in the abstract. We have now defined "Oral Glucose Tolerance Test (OGTT)" at its first mention to improve clarity.
Regarding your suggestion on insulin resistance and molecular mechanisms, we agree that incorporating a brief mention of future studies assessing insulin resistance would strengthen the connection to insulin sensitivity. As a result, we have added the following sentence to the discussion (lines 371-373):
“While we focused on glycemic control and insulin levels, future studies on insulin sensitivity using the Homeostatic Model Assessment of Insulin Resistance (HOMA-IR) could enhance the translational value of these findings.”
We appreciate your attention to detail.
Best regards,
Elena Rafailovska
Reviewer 2 Report
Comments and Suggestions for Authors
They responded to all my comments effectivelly
Author Response
Dear Reviewer,
Thank you for your valuable feedback, which greatly improved our manuscript.
Best regards,
Elena Rafailovska
(On behalf of all authors)
Reviewer 4 Report
Comments and Suggestions for Authors
Author has done significant improvement while revising the manuscript. It can be accepted for further processing.
Author Response

(The authors gave the same response as above.)
